# Using Different Ions in the Hydrothermal Method to Enhance the Photoluminescence Properties of Synthesized ZnO-Based Nanowires

**Ya-Fen Wei [1,2], Wen-Yaw Chung [2,*], Cheng-Fu Yang [1,3,*], Jei-Ru Shen [3] and Chih-Cheng Chen [1]**

[1] School of Information Engineering, Jimei University, Xiamen 361021, China; yafenwei@jmu.edu.cn (Y.-F.W.); 201761000018@jmu.edu.cn (C.-C.C.)

[2] Department of Electronic Engineering, Chung Yuan Christian University, Taoyuan City 320, Taiwan

[3] Department of Chemical and Material Engineering, National University of Kaohsiung, Kaohsiung 811, Taiwan; maltese1114@gmail.com

[*] Correspondence: eldanny@cycu.edu.tw (W.-Y.C.); cfyang@nuk.edu.tw (C.-F.Y.); Tel.: +886-3-2654602 (W.-Y.C.); +886-7-5919283 (C.-F.Y.)

**Abstract:** ZnO films with a thickness of ~200 nm were deposited on $SiO_2$/Si substrates as the seed layer. Then $Zn(NO_3)_2$-$6H_2O$ and $C_6H_{12}N_4$ containing different concentrations of $Eu(NO_3)_2$-$6H_2O$ or $In(NO_3)_2$-$6H_2O$ were used as precursors, and a hydrothermal process was used to synthesize pure ZnO as well as Eu-doped and In-doped ZnO nanowires at different synthesis temperatures. X-ray diffraction (XRD) was used to analyze the crystallization properties of the pure ZnO and the Eu-doped and In-doped ZnO nanowires, and field emission scanning electronic microscopy (FESEM) was used to analyze their surface morphologies. The important novelty in our approach is that the ZnO-based nanowires with different concentrations of $Eu^{3+}$ and $In^{3+}$ ions could be easily synthesized using a hydrothermal process. In addition, the effect of different concentrations of $Eu^{3+}$ and $In^{3+}$ ions on the physical and optical properties of ZnO-based nanowires was well investigated. FESEM observations found that the undoped ZnO nanowires could be grown at 100 °C. The third novelty is that we could synthesize the Eu-doped and In-doped ZnO nanowires at temperatures lower than 100 °C. The temperatures required to grow the Eu-doped and In-doped ZnO nanowires decreased with increasing concentrations of $Eu^{3+}$ and $In^{3+}$ ions. XRD patterns showed that with the addition of $Eu^{3+}$ ($In^{3+}$), the diffraction intensity of the (002) peak slightly increased with the concentration of $Eu^{3+}$ ($In^{3+}$) ions and reached a maximum at 3 (0.4) at%. We show that the concentrations of $Eu^{3+}$ and $In^{3+}$ ions have considerable effects on the synthesis temperatures and photoluminescence properties of $Eu^{3+}$-doped and $In^{3+}$-doped ZnO nanowires.

**Keywords:** ZnO-based nanowires; hydrothermal method; $Eu^{3+}$ and $In^{3+}$ ions; photoluminescence properties

## 1. Introduction

Nanostructured semiconducting ZnO-based materials have been widely investigated, attracting significant attention due to their novel physical and chemical properties. Important applications include solar cells [1], light-emitting diodes [2], and super-hydrophobic surfaces [3]. The electron transport efficiency and mechanisms of semiconducting ZnO-based materials are dependent on surface states closely linked to the surface-to-volume ratio. One-dimensional (1D) and two-dimensional (2D) ZnO nanostructures are of great interest because they possess a large surface-to-volume ratio, enabling them to absorb more test molecules or accept more measured signals on their surface and, thereby, be highly efficient sensors. They are considered promising materials for various sensors because they have high

electrochemical stability, are not toxic, are receptive to doping, and are inexpensive [4]. Many different methods of growing ZnO-based nanostructured materials have been investigated. For example, Lupan et al. used a successive ionic layer adsorption and reaction (SILAR) method to deposit undoped and Sn and Ni co-doped nanostructured ZnO thin films on glass [5]. Niarchos et al. investigated a reliable and low-cost method for large-scale ZnO nanorod production, using an alternative aqueous chemical growth (ACG) low-temperature process to grow ZnO nanorods on patterned Si substrates [6]. Kenanakis et al. used an aqueous solution to thoroughly investigate the growth of highly oriented ZnO nanowires on different substrates [7].

Recently, ZnO-based nanostructured materials, including nanotubes, nanowires (nanorods), and thin films, have been synthesized by various physical and chemical methods and used to fabricate sensors for a variety of applications. For example, Mondal et al. used a ZnO-$SnO_2$ composite material to fabricate a micro-electro–mechanical system (MEMS) microheater on silicon (Si) to develop a low-power gas sensor [8]. Zhao et al. grew a single ZnO nanowire on a flexible substrate using a custom-built nano-manipulation system and investigated it as an ultra-high-sensitivity strain sensor [9]. Thomas et al. used a microwave successive ionic layer adsorption reaction to synthesize pure and Al-doped photosensitive ZnO films and investigated their luminescence properties [10].

Various attempts have been made to grow pure ZnO and different ion-doped ZnO nanowires and investigate them as different applications. For example, Bai et al. used a seed-assisted hydrothermal method to grow the Al-doped ZnO nanowires on a silicon substrate [11]. Even though they could grow the ZnO nanowires at a low temperature of 95 °C, this method is very complicated to grow the Al-doped ZnO nanowires because they needed to deposit the Al on ZnO seed layer and annealed the samples at 550 °C. Chang et al. used a traditional thermal evaporation method to synthesize ZnO nanowires on a (100) Si substrate and then used a molecular beam epitaxy system to subsequently carry out the Mn doping process [12]. They constructed a back-gated Mn-doped ZnO nanowire field-effect transistor (FET) to demonstrate the electric-field control properties of ferromagnetism.

In-doped ZnO nanowires can be synthesized and grown using different methods. For example, the In-doped ZnO nanowires were synthesized via a thermal evaporation process, using metallic powders of zinc and indium as precursors and oxidizing them in the presence of oxygen [13,14]. Xu et al. also used a thermal evaporation method to synthesize the In-doped ZnO nanowires, but the used precursors were ZnO and $In_2O_3$ powders [15]. The Eu-doped ZnO nanowires could also be synthesized using different methods. Rifai and Kulnitskiy used chemical vapor deposition method to synthesize the single-crystal $Eu^{3+}$-doped wurtzite ZnO micro- and nanowires [16]. Lupan et al. investigated Eu-doped ZnO nanowire arrays that could be electrodeposited in a three electrode electrochemical cell using an aqueous solution containing $ZnCl_2$, KCl, and $EuCl_3$ as the supporting electrolyte [17]. Geburt et al. first used the vapor–liquid–solid mechanism to synthesize ZnO nanowires with diameters of about 100 to 300 nm. After that, they used ion implantation to dope the $Eu^{3+}$ ions in ZnO nanowires [18]. Apparently, the concentrations of $Eu^{3+}$ and $In^{3+}$ ions in these researches are difficult to control well, and it is difficult to investigate the effects of different concentrations of $Eu^{3+}$ and $In^{3+}$ ions on their properties.

When all the synthesis methods are compared, using an aqueous solution to grow ZnO nanostructured materials is considered better, mainly due to low growth temperature and good potential for large-scale production. Previously, we found that changing the deposition parameters in our hydrothermal method (the concentrations of $Zn(NO_3)_2$-6$H_2O$ and $C_6H_{12}N_4$; the face direction of ZnO/$SiO_2$/Si substrates during hydrothermal deposition; and deposition time) resulted in ZnO films with three different morphologies on ZnO/$SiO_2$/Si substrates [19]. Different deposition parameters yielded undoped ZnO in the shape of irregular plate-structured films, nanowires (nanorods), and chrysanthemum-like clusters (nanoflower films). We also found that with a synthesis temperature of 100 °C, nanowires could be grown using an undoped ZnO solution. ZnO-based nanowires can be used in many fields, for example, as ultraviolet (UV) photodetectors or sensors [20]. Trivalent lanthanide ion-doped wide-bandgap semiconducting ZnO nanowires could be the promising active

materials in opto-electronic devices [17]. We believe that if the photoluminescence excitation (PLE) and photoluminescence emission (PL) properties are enhanced, the sensitivity of these fabricated UV photodetectors or sensors and the emission properties of opto-electronic devices will also be improved. We, therefore, used $Eu^{3+}$ and $In^{3+}$ ions and investigated the PLE and PL properties of Eu-doped and In-doped ZnO nanowires.

In the present study, the first important novelty is that we used $Eu(NO_3)_3$-$6H_2O$ and $In(NO_3)_2$-$6H_2O$ as the dopant sources of $Eu^{3+}$ and $In^{3+}$ ions because we could control their concentrations well. Next, we could synthesize the Eu-doped and In-doped ZnO nanowires using the hydrothermal method at low temperature (below 100 °C) and investigate the effects of different concentrations of $Eu^{3+}$ and $In^{3+}$ ions on their crystalline and photoluminescence properties. The last novelty is that the $Eu^{3+}$ and $In^{3+}$ ions form a compound with ZnO during the synthesis process of ZnO-based nanowires, for that the $Eu^{3+}$ and $In^{3+}$ ions can reach the whole region of the synthesized Eu-doped and In-doped ZnO nanowires. We found that at 100 °C, ZnO-based nanowires did not grow well when different concentrations of $Eu^{3+}$ and $In^{3+}$ ions were added. The needed synthesis temperatures drop increased with the concentrations of $Eu^{3+}$ and $In^{3+}$ ions. We will demonstrate how the designs of synthesis temperature and precursor materials can control the structures of pure ZnO and of Eu-doped and In-doped ZnO nanowires with hexagonal prismatic structures.

As compared with other researches, another important novelty is that no researches prove tht the concentrations of $Eu^{3+}$ and $In^{3+}$ ions will affect the growth temperatures of Eu-doped and In-doped ZnO nanowires and investigate the effect of the concentrations of $Eu^{3+}$ and $In^{3+}$ ions on the growth morphologies and PL properties. As the hydrothermal method was used to grow the Eu-doped and In-doped ZnO nanowires, their crystal qualities have been enhanced. The visible-light emission, which is caused by the defect states, was not found in Eu-doped and In-doped ZnO nanowires and the near-band-edge emission peak (located at around 395 nm) was enhanced because the number of defect states was reduced. We thoroughly investigate the effects of different concentrations of $Eu^{3+}$ and $In^{3+}$ ions on the growth properties of ZnO-based nanowires. We also show that the ions used and their concentrations have considerable effects on the synthesis temperatures and photoluminescence properties of ZnO-based nanowires.

## 2. Experimental

A ZnO seed layer is necessary to initialize the uniform growth of oriented nanowires using aqueous solutions. ZnO powder (Us Research Nanomaterials Inc., purity 99.99%, particle sizes small than 1 µm, Houston, TX, USA) was mixed with polyvinyl alcohol (PVA, ECHO Chemical Co., Ltd. Miaoli, Taiwan) as a binder, and the ZnO-PVA mixture was pressed into pellets 6 mm thick and 56 mm in diameter using a steel die. After debindering, each ZnO pellet was sintered at 1100 °C for 2 h to form a ceramic target. $SiO_2$/Si (Summit-Tech Resource Corp. Hsinchu, Taiwan) was used as the substrate to fabricate pure (undoped) as well as $Eu^{3+}$-doped and $In^{3+}$-doped ZnO nanowires. First, the $SiO_2$/Si substrates were cleaned with deionized (DI) water (office created), acetone (ECHO Chemical Co., Ltd.), and isopropyl alcohol (ECHO Chemical Co., Ltd.), then radio frequency (RF) magnetron sputtering was used to deposit the ZnO seed layers (ZnO/$SiO_2$/Si substrates). Next, $Zn(NO_3)_2$-$6H_2O$ (Alfa Aesar, MA, USA), $C_6H_{12}N_4$ (ECHO Chemical Co., Ltd.), and $Eu(NO_3)_3$-$6H_2O$ (Alfa Aesar) or $In(NO_3)_2$-$6H_2O$ (Alfa Aesar) were mixed with DI water with the designed compositions. The $Zn(NO_3)_2$-$6H_2O$, $Eu(NO_3)_3$-$6H_2O$, and $In(NO_3)_2$-$6H_2O$ decomposed to form the mixed solutions of $Zn^{2+}$ and $In^{3+}$ ions or $Zn^{2+}$ and $Eu^{3+}$ ions in DI water. It was impossible for $In^{3+}$ and $Eu^{3+}$ ions to be separated with $Zn^{2+}$ ions from the solutions during the growth processes of ZnO-based nanowires, the Eu-doped and In-doped ZnO nanowires could be grown from the mixed solutions. The $Eu^{3+}$-doped and $In^{3+}$-doped ZnO nanowires were synthesized at temperatures of 100 to 60 °C for 1 h using a hydrothermal method. Previously, we used a well-designed structure to grow ZnO-based nanostructured materials via a hydrothermal method [19]. We found that when $Zn(NO_3)_2$-$6H_2O$ and $C_6H_{12}N_4$ were used as reagents to synthesize

ZnO-based nanostructured materials, the concentration of the diluted solution had important effects on the synthesis results. We, therefore, fixed the concentration of the diluted solution as a reference point [19].

We also found that the face direction of the ZnO seed layer and the synthesis time were two important factors affecting the synthesis of ZnO-based nanostructured materials. We found that when the face direction was down, nanowires grew on the ZnO seed layer, and when the direction was up, nanoflowers grew on the layer. We also determined that 1 h was the best synthesis duration for growing ZnO nanowires on the ZnO seed layer. Hence, facedown and 1 h were used as the parameters for growing pure, $Eu^{3+}$-doped, and $In^{3+}$-doped ZnO nanowires. The compositions for growing the $Eu^{3+}$-doped nanowires were ZnO + y $Eu^{3+}$ ions, where y = 0, 1, 2, or 3 at%, abbreviated as ZnO-0-Eu (undoped-ZnO), ZnO-10-Eu, ZnO-20-Eu, ZnO-30-Eu, and ZnO-40-Eu. The compositions for growing the $In^{3+}$-doped ZnO nanowires were ZnO + x $In^{3+}$ ions, where x = 0, 0.4, 0.8, or 1.2 at%, abbreviated as ZnO-4-In, ZnO-8-In, and ZnO-12-In. The morphologies of pure ZnO and $Eu^{3+}$-doped and $In^{3+}$-doped ZnO nanowires were observed by field-emission scanning electron microscopy (FESEM, Hitachi 4800, Tokyo, Japan) and used to determine the effects of synthesis temperature and the concentrations of $Eu^{3+}$ and $In^{3+}$ ions on the synthesis properties of the pure and doped ZnO nanowires. Crystalline phases were analyzed using X-ray diffraction (XRD, D8, Bruker, MA, USA) to determine the effects of concentrations of $Eu^{3+}$ and $In^{3+}$ ions on the crystalline and photoluminescence properties of the ZnO-based nanowires.

## 3. Results

In the case of undoped ZnO grown at 80 °C, ZnO nanowires were not synthesized on the $ZnO/SiO_2/Si$ substrate and only irregular ZnO nano-particles were observed, as Figure 1a shows (presents a top-down image). As the temperature was 100 °C, ZnO nanowires were successfully synthesized on the $ZnO/SiO_2/Si$ substrate, as Figure 1b shows. To investigate the effects of concentrations of $Eu^{3+}$ and $In^{3+}$ ions on the synthesis properties of hydrothermally grown $Eu^{3+}$-doped and $In^{3+}$-doped ZnO nanostructures, the synthesis temperature of all the ZnO + y $Eu^{3+}$ and ZnO + x $In^{3+}$ compositions was set at 100 °C. The general surface morphologies of the hydrothermally grown $Eu^{3+}$-doped ZnO nanostructures were examined by FESEM, and the results are also shown in Figure 1. The SEM image of this sample revealed that the undoped ZnO nanowires had a hexagonal wurtzite structure, with an easily discernable hexagonal prism arrangement. When the compositions were changed to ZnO-20-Eu and ZnO-40-Eu, as Figure 1c,d show, no nanowires grew on the $ZnO/SiO_2/Si$ substrates and only irregular plate-structured grains were observed. When the compositions were changed to ZnO-60-Eu and ZnO-80-Eu (not shown here), only very large, irregular plate-structured grains were observed. These results prove that synthesis temperature is an important parameter to affect the synthesized results of ZnO-based nanowires.

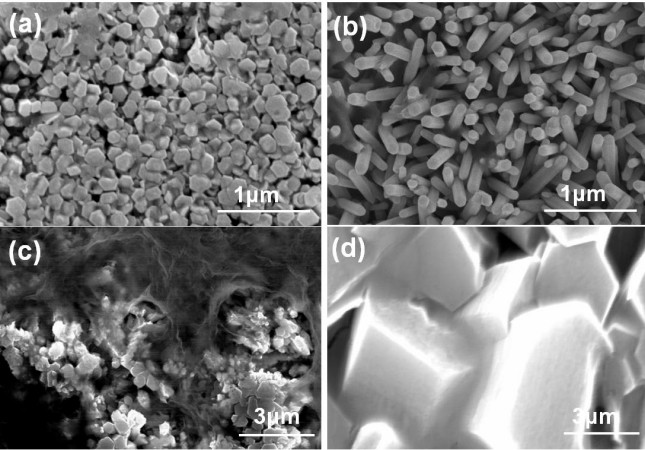

**Figure 1.** Surface morphologies of hydrothermally grown $Eu^{3+}$-doped ZnO nanostructures, (**a**) undoped ZnO synthesized at 80 °C; synthesized at 100 °C: (**b**) undoped ZnO, (**c**) ZnO-20-Eu, and (**d**) ZnO-40-Eu.

However, when $In^{3+}$ was used, the surface morphologies of the $In^{3+}$-doped ZnO nanostructures resembled those of the $Eu^{3+}$-doped ZnO nanostructures. When ZnO-4-In, ZnO-8-In, and ZnO-12-In were used to grow ZnO-based nanowires at 100 °C, their FESEM images were similar to those of ZnO-20-Eu and ZnO-40-Eu: no nanowires grew, and only irregular plate-structured grains were observed (not shown here). These results suggest that when the ZnO seed layer is deposited, the concentrations of $Eu^{3+}$ or $In^{3+}$ ions have an important effect on the synthesis results of the ZnO-based nanostructures. Increasing the concentrations of $Eu^{3+}$ or $In^{3+}$ ions lowers the temperature required to form the ZnO-based nanowires. We next demonstrated that the concentrations of $Eu^{3+}$ or $In^{3+}$ ions (or the synthesis temperature) are the most important factor in the syntheses of $Eu^{3+}$-doped and $In^{3+}$-doped ZnO nanowires on ZnO/SiO$_2$/Si substrates.

The $Eu^{3+}$-doped ZnO nanomaterials were synthesized at 100 °C using various concentrations of $Eu^{3+}$ ions. The XRD patterns of the $Eu^{3+}$-doped ZnO nanostructures are shown in Figure 2. Only the diffraction peak of the (002) plane was observed; no (004) plane diffraction peak appeared. Apparently, the diffraction intensity of the $Eu^{3+}$-doped ZnO nanostructures decreased as the concentration of $Eu^{3+}$ ions increased, reaching a minimum at ZnO-30-Eu, then becoming saturated as the concentration of $Eu^{3+}$ ions was further increased. Using Figure 2, we also measured the 2θ value and full width at half maximum (FWHM) value of the (002) plane of the $Eu^{3+}$-doped ZnO nanostructures. The 2θ values of the (002) plane of the 100 °C-synthesized $Eu^{3+}$-doped ZnO nanostructures were unchanged at 34.44 ± 0.02 as the concentration of $Eu^{3+}$ ions increased. The FWHM value of the (002) plane initially increased, reaching saturation when the concentration of $Eu^{3+}$ ions was 3 at%. These results suggest that the crystallinity of the $Eu^{3+}$-doped ZnO nanostructures degenerated as the concentration of $Eu^{3+}$ ions increased. Comparison with the results in Figure 1 leads us to believe that the degeneration was caused by unformed nanostructures.

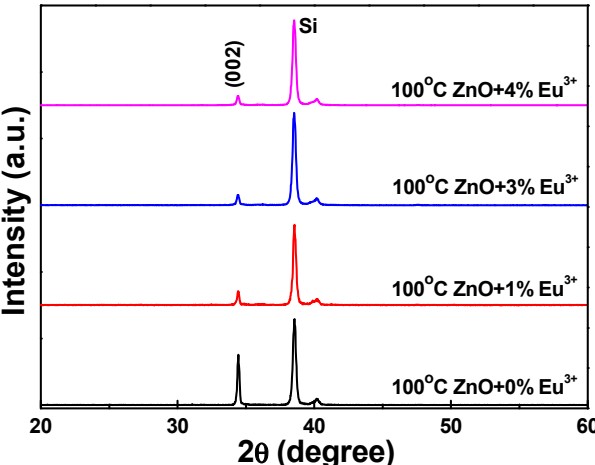

**Figure 2.** X-ray diffraction (XRD) patterns of synthesized $Eu^{3+}$-doped ZnO nanowires as a function of the concentration of $Eu^{3+}$ ions; the synthesis temperature was 100 °C.

When $In^{3+}$ was used as the ion, the XRD patterns of the $In^{3+}$-doped ZnO nanomaterials were similar to those of the $Eu^{3+}$-doped ZnO nanomaterials (not shown here). These results also suggest that increasing the concentration of $In^{3+}$ ions led the $In^{3+}$-doped ZnO to form ZnO-based nanowires at lower temperatures. In addition, only the diffraction peak of the (002) plane was observed in the $In^{3+}$-doped ZnO nanostructures; no diffraction peak for the (004) plane was evident. The diffraction intensity of the $In^{3+}$-doped ZnO nanostructures decreased as the concentration of $In^{3+}$ ions increased, reaching a minimum at ZnO-4-In, then it was unchanged as the concentration of $In^{3+}$ ions was further increased.

The morphologies of the synthesized $Eu^{3+}$-doped ZnO nanowires are shown in Figure 3 for different concentrations of $Eu^{3+}$ ions and synthesis temperatures. Figure 3a shows that when ZnO-10-Eu was used at 90 °C, nanowires with diameters in the range of 50 to 160 nm and hexagonal prism

structures were readily observable. When the concentration of $Eu^{3+}$ ions increased, the synthesis temperature could be lowered. As Figure 3b–d show, the synthesis temperatures of the ZnO-20-Eu, ZnO-30-Eu, and ZnO-40-Eu nanowires were 80, 70, and 60 °C, and their diameters were in the ranges of 50 to 85, 40 to 80, and 140 to 450 nm, respectively. Other than the nanowire structure changing from an equilateral hexagon to a non-equilateral hexagon, the surface morphologies experienced no apparent change, and all had a hexagonal prism structure.

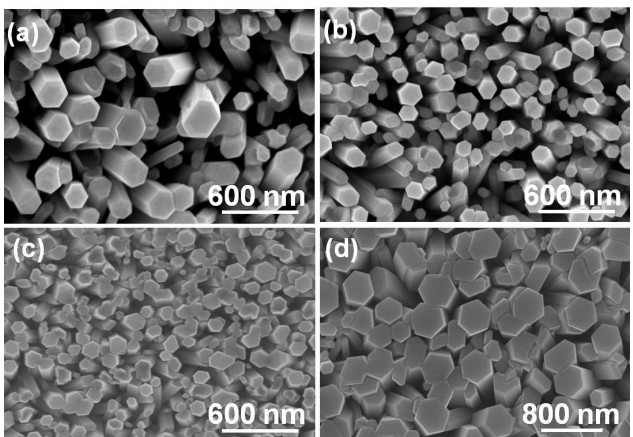

**Figure 3.** Surface morphologies of hydrothermally grown $Eu^{3+}$-doped ZnO. (**a**) ZnO-10-Eu nanowires synthesized at 90 °C, (**b**) ZnO-20-Eu nanowires synthesized at 80 °C, (**c**) ZnO-30-Eu nanowires synthesized at 70 °C, and (**d**) ZnO-40-Eu nanowires synthesized at 60 °C.

Figure 4 shows top-view SEM images of high-density $In^{3+}$-doped ZnO nanowires grown on $ZnO/SiO_2/Si$ substrates at different synthesis temperatures and with different concentrations of $In^{3+}$ ions. Figure 4a shows that the 88 °C ZnO-4-In nanowires had the structure of hexagonal prisms with diameters in the range of 45 to 150 nm. When the concentration of $In^{3+}$ ions was increased, the $In^{3+}$-doped ZnO nanowires could be synthesized at a low temperature. As Figure 4b,c shows, the synthesis temperatures of ZnO-8-In and ZnO-12-In nanowires were 75 and 60 °C, and their diameters were in the ranges of 130 to 280 and 70 to 150 nm, respectively. The top-view image shows that the $In^{3+}$-doped ZnO nanowires changed from an equilateral hexagon structure to a non-equilateral hexagon structure as the concentration of $In^{3+}$ ions increased. When $Eu^{3+}$ or $In^{3+}$ ions were added, the nanowires still displayed a hexagonal wurtzite structure, providing strong evidence that the undoped ZnO and the $Eu^{3+}$-doped and $In^{3+}$-doped ZnO nanowires grew in the (002) direction, independent of the concentrations of $Eu^{3+}$ or $In^{3+}$ ions.

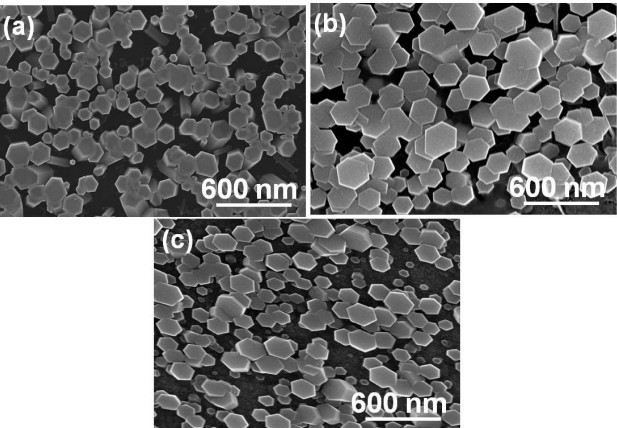

**Figure 4.** Surface morphologies of hydrothermally grown $In^{3+}$-doped ZnO: (**a**) ZnO-4-In nanowires synthesized at 88 °C, (**b**) ZnO-8-In nanowires synthesized at 75 °C, and (**c**) ZnO-12-In nanowires synthesized at 60 °C.

Table 1 shows the corresponding FESEM images of formed Eu-doped and In-doped ZnO nanowires (Figures 3 and 4) the elemental ratios obtained by FESEM equipped with energy dispersive X-ray spectroscopy (EDX) analyses for elemental Zn and Eu or Zn and In. Six different areas of Eu-doped and In-doped ZnO nanowires were depicted for analysis, and the range of the measured elemental ratio for Eu and In and the average value of the measured elemental ratio are shown in Table 1. The Eu and In elements were detected in Eu-doped and In-doped ZnO nanowires, respectively, and the measured elemental ratios increased with the concentrations of $Eu^{3+}$ and $In^{3+}$ ions.

**Table 1.** Energy dispersive X-ray spectroscopy (EDX) analyses for elemental Zn and Eu or Zn and In. Eu(In)-measured: the range of the measured elemental ratio; Eu(In)-average: the average value of the measured elemental ratio.

| Composition | Eu-Measured | Eu-Average | Composition | In-Measured | In-Average |
|---|---|---|---|---|---|
| ZnO-10-Eu | 0.56–0.78% | 0.69% | ZnO-4-In | 0.04–0.34% | 0.21% |
| ZnO-20-Eu | 1.32–1.77% | 1.47% | ZnO-8-In | 0.38–0.72% | 0.59% |
| ZnO-30-Eu | 2.02–2.51% | 2.24% | ZnO-12-In | 0.85–1.11% | 1.02% |
| ZnO-40-Eu | 3.25–3.56% | 3.32% | | | |

We investigated the crystallinity of $Eu^{3+}$-doped ZnO nanowires synthesized with different concentrations of $Eu^{3+}$ ions and synthesis temperatures using XRD, and the results are shown in Figure 5. All of the patterns were in agreement with the diffraction data from the standard card (JCPDS 36-1451). The main diffraction peak of ZnO is (101) (JCPDS 36-1451), which is located around 2θ~36.25°. However, the stronger intensity of the (002) diffraction peak, which is located at 2θ = 34.44 ± 0.02~34.40 ± 0.02, was discernible for all of the $Eu^{3+}$-doped ZnO nanowires, suggesting that all of the $Eu^{3+}$-doped ZnO nanowires had a high c-axis orientation. The 2θ value of the c-orientation (200) peak decreased from 34.43 ± 0.02, 34.43 ± 0.02, 34.42 ± 0.02, 34.41 ± 0.02, to 34.40 ± 0.02 as the concentration of $Eu^{3+}$ ions increased from 0 1, 2, 3, to 4 at%. The radius of $Eu^{3+}$ ions larger than that of $Zn^{2+}$ ions is the reason to cause an unapparent decrease in the 2θ value of the c-orientation (200) peak. The FWHM value of the (200) diffraction peak decreased from 2θ = 0.19 (34.37–34.55), 0.18 (34.35–34.52), 0.17 (34.34–34.50), to 0.15 (34.34–34.48) as the concentration of $Eu^{3+}$ ions increased from 0, 1, 2, to 3 at%, indicating that the crystallization of the nanowires increased with the concentration of $Eu^{3+}$ ions. When the concentration of $Eu^{3+}$ ions increased from 3 or 4 at%, the (100) peak was observed, the diffraction intensity of the (200) diffraction peak increased, and the FWHM value increased from 2θ = 0.15 to 0.17 (34.34–34.50). Figure 5 also shows that the diffraction intensity of the (100) peak increased and the FWHM value decreased as the concentration of $Eu^{3+}$ ions increased from 3 to 4 at%. These results suggest that the crystallization property changed when the concentration of $Eu^{3+}$ ions was 3 at% or more.

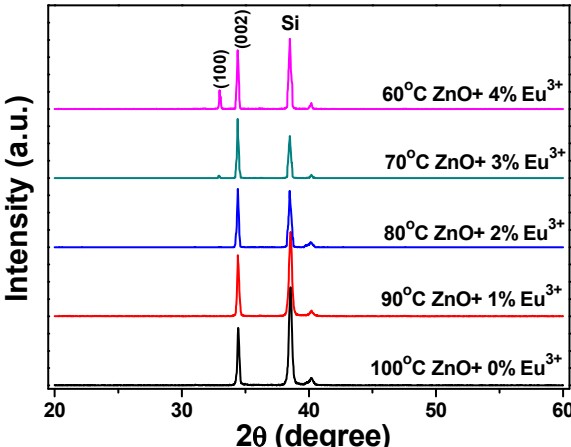

**Figure 5.** XRD patterns of the synthesized $Eu^{3+}$-doped ZnO nanowires as a function of the concentration of $Eu^{3+}$ ions (or synthesis temperature).

The crystallinities of In$^{3+}$-doped ZnO nanowires synthesized with different concentrations of Eu$^{3+}$ ions were investigated using XRD, and the results are shown in Figure 6. Comparison of these results showed differences when the concentration of In$^{3+}$ ions was varied. As the concentration of In$^{3+}$ ions was increased from 0 to 0.4 at%, the FWHM value of the (200) diffraction peak decreased from 2θ = 0.18 (34.37–34.55) to 0.17 (34.37–34.54) and the diffraction intensity increased. As the concentration of In$^{3+}$ ions was further increased from 0.4, 0.8, to 1.2 at%, the FWHM value of the (200) diffraction peak decreased from 2θ = 0.17, 0.20 (34.36–34.55), to 0.25 (34.33–34.57) and the diffraction intensity decreased. Nevertheless, the synthesized In$^{3+}$-doped ZnO nanowires exhibited no (100) diffraction peak. Further analysis of the XRD data in Figure 6 showed that the radius of the In$^{3+}$ ions (0.80 nm) was larger than that of the Zn$^{2+}$ ions (0.74 nm), and all the (002) diffraction peaks were located at 2θ = 34.44 ± 0.02 as the concentrations of In$^{3+}$ ions were 0.0 and 0.4 at% and located at 2θ = 34.43 ± 0.02 as the concentrations of In$^{3+}$ ions were 0.8 and 1.2 at%. Even the radius of In$^{3+}$ ions is larger than that of Zn$^{2+}$ ions, but the concentration of In$^{3+}$ ions used to dope into the In-doped ZnO nanowires was very low, and the 2θ value of the c-orientation (200) peak was almost unchanged. The results in Figures 1–6 prove that when the undoped ZnO and the Eu$^{3+}$-doped ZnO and In$^{3+}$-doped ZnO nanowires grew well, the synthesized ZnO-based nanowires had a good wurtzite hexagonal crystal structure. The undoped ZnO, the Eu$^{3+}$-doped ZnO, and the In$^{3+}$-doped ZnO had similar crystalline results; thus, we believe they will have different PLE and PL spectra.

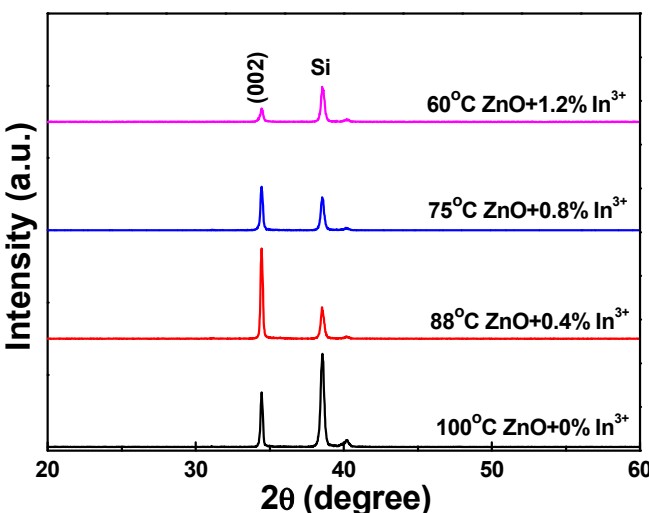

**Figure 6.** XRD patterns of the synthesized In$^{3+}$-doped ZnO nanowires as a function of the concentration of In$^{3+}$ ions (or synthesis temperature).

The radii of 4-coordination and 6-coordination Zn$^{2+}$ ions are 60 and 74 pm, and the radii of 4-coordination and 6-coordination In$^{3+}$ ions are 62 and 80 pm; 4-coordination Eu$^{3+}$ ions do not exist, and the radius of 6-coordination Eu$^{3+}$ ions is 95 pm. The values of the calculated lattice constant, *c*, of the Eu$^{3+}$-doped and In$^{3+}$-doped ZnO nanowires were considerably smaller than the *c* value of the undoped ZnO nanowires. Because all the investigated ZnO-based nanowires had a hexagonal wurtzite structure, the Zn$^{2+}$, Eu$^{3+}$, and In$^{3+}$ ions were in a 6-coordination structure. Comparison of the results in Figures 2, 5 and 6 shows that the radii of Eu$^{3+}$ (0.95 nm) and In$^{3+}$ (0.80 nm) are larger than that of Zn$^{2+}$ (0.74 nm), and no variation in the lattice parameters of the undoped, Eu$^{3+}$-doped, and In$^{3+}$-doped ZnO nanowires was observable. However, comparison of the images in Figures 1a, 3 and 4 show that from the top view, the ZnO-based nanowires changed from an equilateral hexagon to a non-equilateral hexagon configuration as the concentrations of Eu$^{3+}$ and In$^{3+}$ ions increased. We believe this was the result of the differences between the radius of Zn$^{2+}$ ions and the radius of Eu$^{3+}$ ions.

Figure 7 shows the room-temperature PLE spectra of different ion-doped ZnO nanowires recorded in the wavelength range of 200 to 350 nm. The PLE spectra were measured at an emission wavelength

of 395 nm, while the PL spectra were measured for maximum emission intensity. There are obvious differences between the PLE spectra of pure ZnO and different ion-doped ZnO nanowires. The emission intensity of the PLE spectra of the $Eu^{3+}$-doped ZnO nanowires increased with the concentration of $Eu^{3+}$ ions and reached a maximum value in the 60 °C-grown ZnO-40-Eu, and that of $In^{3+}$-doped ZnO nanowires appeared in the 88 °C-grown ZnO-4-In. Comparison of the results in Figures 1–4 suggests that the photoluminescence properties of the ZnO-based nanowires were dependent on their crystalline properties. We, therefore, used the PLE and PL spectra of the 100 °C-grown ZnO, 60 °C-grown ZnO-40-Eu, and 88 °C-grown ZnO-4-In nanowires to compare their optical properties.

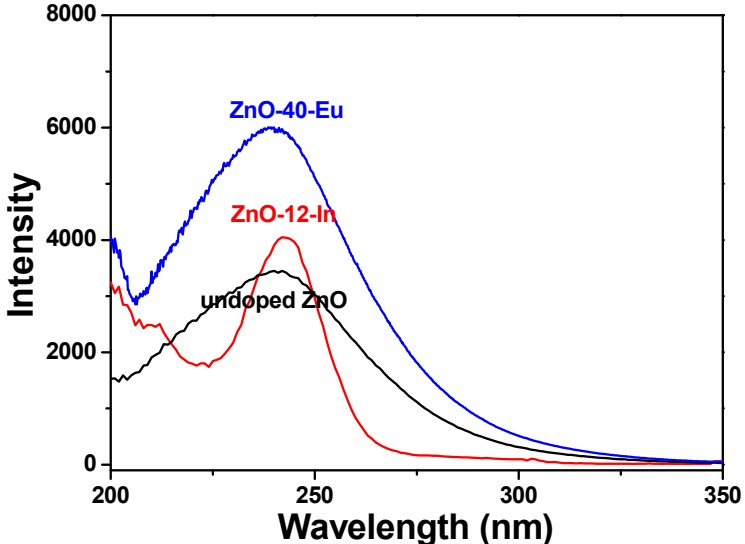

**Figure 7.** Photoluminescence excitation (PLE) spectra of 100 °C-grown ZnO, 60 °C-grown ZnO-40-Eu, and 88 °C-grown ZnO-4-In nanowires.

The PLE spectra in Figure 7 changed as different ions were added. As the wavelength in Figure 7 increased from 200 to 350 nm, the PLE emission intensities first decreased to a minimum, then increased to a maximum, then decreased again. Compared with that of pure ZnO nanowires, the wavelengths of 0 °C-grown ZnO-40-Eu, and 88 °C-grown ZnO-4-In nanowires reveal the minimum intensities were shifted to higher values and wavelengths of them reveal the maximum had no apparent changes. When different ions were added to synthesize the ZnO-based nanowires, they caused dissimilar energy level transitions in different energy bands, leading the phosphors to release light with different emission peaks (or wavelengths). This is why the PLE spectra of the 100 °C-grown ZnO, 60 °C-grown ZnO-40-Eu, and 88 °C-grown ZnO-4-In nanowires differed. The broadening of the emission peaks of the 100 °C-grown ZnO and 60 °C-grown ZnO-40-Eu nanowires can be interpreted by the formation of band tailing in the band gap, which often is induced by the formation of the defects in the semiconductor during the synthesis process or the introduction of an impurity into the semiconductor [15].

The room-temperature PL emission spectra of the 100 °C-grown ZnO, 60 °C-grown ZnO-40-Eu, and 88 °C-grown ZnO-4-In nanowires were excited using a wavelength of 242 nm and recorded in the range of 200 to 700 nm. These PL properties were obtained at room temperature, and the results are compared in Figure 8. In addition, there are obvious differences between the PL spectra of pure ZnO and different ion-doped ZnO nanowires. The wavelengths resulting in the maximum emission intensities were 395 ± 2, 396± 2, and 397 ± 2 nm for 100 °C-grown undoped ZnO, 60 °C-grown ZnO-40-Eu, and 88 °C-grown ZnO-4-In nanowires, respectively. These results suggest that the additions of $Eu^{3+}$ or $In^{3+}$ affect the maximum intensities of the emission spectra, but they almost cannot affect wavelengths resulting in the maximum emission intensities. The 100 °C-grown undoped ZnO and 60 °C-grown ZnO-40-Eu nanowires had two distinct, visible-light emission peaks, one centered at ~361 nm and the other at 395 or 396 nm, and the emission spectra ranged from 280 to 570 nm.

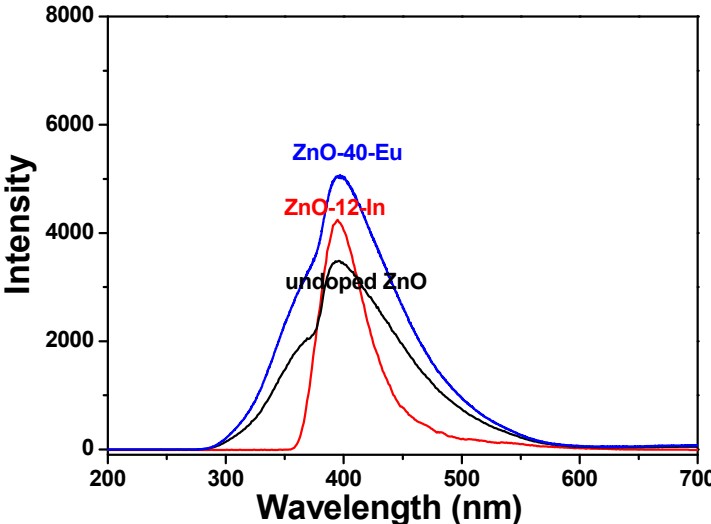

**Figure 8.** Photoluminescence emission (PL) spectra of 100 °C-grown ZnO, 60 °C-grown ZnO-40-Eu, and 88 °C-grown ZnO-4-In nanowires.

With the 88 °C-grown ZnO-4-In nanowires, only one peak was observed, and the emission spectra ranged from 360 to 520 nm. The maximum intensities of the PL emission spectra were 100 °C-grown ZnO < 88 °C-grown ZnO-4-In < 60 °C-grown ZnO-40-Eu. Our results suggest two important findings. First, when a hydrothermal method is used to grow ZnO-based nanowires, adding different concentrations of $Eu^{3+}$ and $In^{3+}$ ions can reduce the synthesis temperature, and this temperature decreases with the concentrations of $Eu^{3+}$ and $In^{3+}$ ions. Second, when different ions in different concentrations are added to grow ZnO-based nanowires, the optical properties of the synthesized ZnO-based nanowires can be controlled. The PLE and PL properties of these ZnO-based nanowires depend on their crystalline properties.

In the past, different researches or different methods to synthesize the undoped and ion-doped ZnO nanowires would have different emission results of PL spectra. Xu et al. measured a PL spectrum using a He-Cd laser of 260 nm as the excitation source at room temperature. They found that the undoped ZnO nanowires has a visible-light emission band with center at about 490 nm and the In-doped ZnO nanowires with an emission band centering at around 398 nm dominate in the PL spectrum [15]. Kim et al. used He-Cd (325 nm) laser as the excitation source, they found that the room-temperature PL spectroscopy of In-doped ZnO nanowires exhibits an unapparent UV emission centered at 378 nm and a broad emission centered at 561 nm [13]. Zeng et al. found that as a Xe lamp was used as the excitation light, the room-temperature PL properties of an undoped ZnO nanosheet featured a broad yellow band centered at 575 nm and they thought that the emission of yellow light is attributed to transitions of oxygen interstitials (Oi) [21]. They also found that as UV (464 nm) light was used as the excitation light, the spectra of Eu-doped ZnO nanowires emitted a pure red luminescence and consisted of a series of resolved emission peaks centered at 577, 589, 612, 619, and 654 nm, they can be assigned to the transitions of $^5D_0{\rightarrow}^7F_0$ (577 nm), $^5D_0{\rightarrow}^7F_1$ (589 nm), $^5D_0{\rightarrow}^7F_2$ (612, 619 nm), and $^5D_0{\rightarrow}^7F_3$ (654 nm), respectively [22].

Geburt et al. also found that the Eu-doped ZnO nanowires have the sharp and structured emission features between 1.5 and 2.1 eV, and these peaks can be clearly assigned to the $^5D_0{\rightarrow}^7F_J$ (J = 0, 1, 2, 3, 4) transitions of $Eu^{3+}$ ions [18]. Lupan et al. recorded the PL spectrum of Eu-doped ZnO nanowires under UV 266 nm excitation, no red emission due to the transitions of $Eu^{3+}$ ions could be detected. The main emission peak centered at 382 nm was found and a weak green luminescence due to ZnO intrinsic defects could also be observed at about 530 nm [17]. Gomi et al. found earlier that interstitial Zn ($Zn_i$) causes the red emission [23] and Teke et al. found that vacancy Zn and oxygen ($V_{Zn}$ and $V_O$) cause the green emission [24]. TekeXing et al. also found that $V_{Zn}$ and $V_O$ cause the green emission and

excess oxygen cause the orange–red emission [25]. Chen et al. also found that the excess oxygen on the surfaces of ZnO nanowires causes the orange–red emission, and it is adjustable via the annealing process or the surfaces' modification of ZnO nanowires [26]. However, we had measured the emission spectrum of ZnO NPs excited by UV light with wavelength of 242 nm, and the main peak was at about 393 nm [27]. Usually, the PL emission of ZnO-based materials is attributed to different defects, such as oxygen vacancies ($V_O$), zinc vacancies ($V_{Zn}$), or is caused by the complex defects of involving interstitial zinc ($Zn_i$) and interstitial oxygen ($O_i$) [28]. The energy level of $Zn_i$ is one type of shallow donor level below conduction band, and it locates 3.15 or 2.91 eV, whereas, the energies of $V_{Zn}$, $O_i$, and $O_{Zn}$ are more acceptors level above the valence band. The 3.15 eV related to light wavelength can be calculated using 1240 (nm)/3.15 = 394 nm. In this study, the centered wavelengths of PL spectra for the grown undoped, Eu-doped, and In-doped ZnO nanowires are about 395 nm. These results suggest that $Zn_i$ is the main reason to cause PL properties of the grown undoped, Eu-doped, and In-doped ZnO nanowires.

Thus, in our case, we used the hydrothermal method to synthesize the ZnO-based nanowires and used $Eu(NO_3)_2 \cdot 6H_2O$ or $In(NO_3)_2 \cdot 6H_2O$ as precursors of the dopants. We believe that the $Eu^{3+}$ and $In^{3+}$ ions will purely occupy the sites of Zn and act as a dopant to change the semiconducting characteristic and enhance the emission intensities of Eu-doped and In-doped ZnO nanowires, and the defects of $Zn_i$, $V_{Zn}$, and $V_O$ will not happen in the synthesized ZnO-based nanowires. The emission peaks of 100 °C-grown undoped ZnO, 60 °C-grown ZnO-40-Eu, and 88 °C-grown ZnO-4-In nanowires centered at about 395 nm are characteristic of ZnO near band edge recombination. However, as the $Eu(NO_3)_2 \cdot 6H_2O$ or $In(NO_3)_2 \cdot 6H_2O$ are used as the dopants of ZnO-based nanowires and the hydrothermal method is used to synthesize the Eu-doped and In-doped ZnO nanowires, their emission intensities of PL spectra are really enhanced. When In is used as dopant in ZnO nanowires, an enhancement in near-band-edge emission peak (located at around 395 nm) occurs than for pure ZnO nanowires while visible-light emission is decreased significantly owing to change in the growth kinetics due to In supply which helps to reduce the number of defect states resulting in improved crystal quality [29]. XRD patterns in Figures 5 and 6 show that the 100 °C-grown undoped ZnO nanowires had high crystal quality and the crystal qualities of the 60 °C-grown ZnO-40-Eu and 88 °C-grown ZnO-4-In nanowires were higher than that of 100 °C-grown undoped ZnO nanowires. For that, the visible-light emission, which is caused by the number of defect states, cannot be found in Figure 8 and only near-band-edge emission peak is observed. We thought that it is the reason that the maximum PL intensities of the 60 °C-grown ZnO-40-Eu and 88 °C-grown ZnO-4-In nanowires are higher than that of the 100 °C-grown undoped ZnO nanowires. The synthesized 60 °C-grown ZnO-40-Eu and 88 °C-grown ZnO-4-In nanowires can be the promising active materials in UV detectors and opto-electronic devices.

## 4. Conclusions

In this study, we used a hydrothermal method to investigate a simple process for synthesizing the $Eu^{3+}$-doped and $In^{3+}$-doped ZnO nanowires at temperatures lower than 100 °C. We found that the requisite synthesis temperatures for undoped ZnO, ZnO-10-Eu, ZnO-20-Eu, ZnO-30-Eu, and ZnO-40-Eu nanowires were 100, 90, 80, 70, and 60 °C, and for ZnO-4-In, ZnO-8-In, and ZnO-12-In nanowires were 88, 75, and 60 °C, respectively. For the undoped, $Eu^{3+}$-doped, and $In^{3+}$-doped ZnO nanowires, the (200) peak was the main diffraction peak, and the 2θ values of the *c*-orientation (200) peak were almost unchanged as the concentration of $Eu^{3+}$ ions increased from 0 to 4 at% and the concentration of $In^{3+}$ ions increased from 0 to 1.2 at%. The (200) diffraction peak was observed when the concentrations of $Eu^{3+}$ ions were 3 and 4 at%; its diffraction intensity increased, and the FWHM value of the (200) diffraction peak decreased with the concentration of $Eu^{3+}$ ions. Two distinct emission peaks were discernible in the 100 °C-grown undoped ZnO and 60 °C-grown ZnO-40-Eu nanowires, one centered at ~300 nm and the other at 395 or 396 nm, and the emission spectra ranged from 280 to 570 nm. The visible-light emission the visible-light emission was not found in the 100 °C-grown undoped-ZnO, 60 °C-grown ZnO-40-Eu, and 88 °C-grown ZnO-4-In nanowires and only near-band-edge emission

peak was observed because their crystal qualities were enhanced, and the number of defect states was reduced. The 88 °C-grown ZnO-4-In nanowires exhibited only one peak, and the emission spectra ranged from 360 to 520 nm. The PL emission spectra showed that the maximum intensities increase in the order 100 °C-grown ZnO < 88 °C-grown ZnO-4-In < 60 °C-grown ZnO-40-Eu.

**Author Contributions:** Y.-F.W. organized the paper and helped with writing it; W.-Y.C. and C.-F.Y. organized the paper and helped with editing the English; J.-R.S. and C.-C.C. carried out the experiments, analyzed the data, and measurements.

**Acknowledgments:** Financial support from the NSFC project titled "Fabrication of high efficiency ammonia gas sensors by using P3HT-zinc oxide nanowire hetero-junction for pilot detection liver cancer" is deeply appreciated.

**Conflicts of Interest:** The authors declare no conflict of interest.

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
