# Peer review of "Using Different Ions in the Hydrothermal Method to Enhance the Photoluminescence Properties of Synthesized ZnO-Based Nanowires"

_electronics, doi:10.3390/electronics8040446_

Round 1

Reviewer 1 Report

The authors prepared ZnO films approx. 200 nm on SiO2/Si substrate by using the hydrothermal method. It was shown that doping of ZnO with Eu(In) enhanced significantly the photpluminescence properties in relation with reference ZnO. The experimental rewsults are well documented.

Author Response

The authors prepared ZnO films approx. 200 nm on SiO2/Si substrate by using the hydrothermal method. It was shown that doping of ZnO with Eu(In) enhanced significantly the photoluminescence properties in relation with reference ZnO. The experimental results are well documented

Answer: Thanks for reviewer’s agreement to our research.

Reviewer 2 Report

The authors address on the growth and characterization of Eu and In doped ZnO nanowires. In the manuscript title it is highlighted that their growth method enhances the luminescence properties of the ZnO nanowires, however, in my opinion, their experiments do not evidence this fact. The originality of the work is low since there are hundreds of papers regarding this issue in the literature.  

Although the synthesis method could be interesting since it allows to growth ZnO nanowires at low temperatures, the nanowires characterization has not any novelty and do not support some of the conclusions.

First of all, there are no evidences of incorporation of the dopants into de nanowires. Neither the XRD nor the PL experiments show any feature indicating this fact.

 Normally, when trivalent ions are introduced into the wurtzite structure, some changes in the lattice parameters are observed since the dopants ions are bigger or smaller and there is a charge imbalance that has to be overcome producing defects and distortions in the structure. The peaks in the XRD do not shift in any of the samples, no matter if the dopant ion is Eu or In and the amount of it. The authors do not give any indication of the value of the lattice parameters. They describe changes in the FWHM and intensity of the peaks but no specific data are given and it is not possible to see these changes from the figures they show. They also claim that since in some of the FESEM images the nanowires seem to be not regular hexagons the dopants have to be incorporated but, in my opinion, this is not enough reason for this conclusion. In fact, how many FESEM images have they taken? Have they made any statistics of this change of shape? In particular in figure 4, part c, it seems to me that the shape change is an artifact of the image due to thermal drift during image acquisition. No compositional analysis of any type has been performed, so it is not clear that any of the dopants are in fact incorporated into the ZnO nanowires.

Regarding the luminescence experiments, they are poorly explained and discussed. There is no explanation of why a wavelength of 393 nm has been used for PLE spectra. The authors do not really explain why there are differences between the spectrum performed on undoped and doped samples. In the PL spectra, they claim that there is a change from 395 to 396 and 397 in the emission peak position, but  1 nm resolution is under the possibilities of the technique at room temperature. In addition, the results discussion is limited to a list of different results of various papers and no comparison with their results is made. It is surprising that no emissions related to Eu are observed since this is one of the most efficient luminescent ions.

In summary, in my opinion, this manuscript is not adequate for publication in its present form and substantial changes should be done to improve its quality, so that my recommendation is to reject it.

Author Response

Reviewer 2:

The authors address on the growth and characterization of Eu and In doped ZnO nanowires. In the manuscript title it is highlighted that their growth method enhances the luminescence properties of the ZnO nanowires, however, in my opinion, their experiments do not evidence this fact. The originality of the work is low since there are hundreds of papers regarding this issue in the literature. Although the synthesis method could be interesting since it allows to growth ZnO nanowires at low temperatures, the nanowires characterization has not any novelty and do not support some of the conclusions.

Answer: Thanks for reviewer’s comment. As reviewer said “there are hundreds of papers regarding this issue”, but I would be like to say that there are no any researches using the hydrothermal method at low temperature (below 100oC) to grow the Eu-doped and In-doped ZnO nanowires. References 13-18, 26, and 28-29 are the relative papers for In- or Eu-doped ZnO nanowires, but these methods are complicated. The important novelties of this research is that we can control the doping concentrations of In3+ and Eu3+ ions in the ZnO nanowires and we can investigate the effect of different concentrations of In3+ and Eu3+ ions on growth characteristics of the ZnO-based nanowires. These technologies cannot be reached in the references 13-18, 26, and 28-29. Also, we added another important novelty, please see lines 113-118 and lines 431-434.

First of all, there are no evidences of incorporation of the dopants into nanowires. Neither the XRD nor the PL experiments show any feature indicating this fact.

Answer: Thanks for reviewer’s comment. References 13-18, 26, and 28-29 are the relative papers for In- or Eu-doped ZnO nanowires, these methods are complicated and they also cannot have any evidences that the In3+ and Eu3+ ions can dope into the nanowires. We added the following description to explain the dopants into nanowires. Please see lines 130-135.

  The Zn(NO3)2-6H2O, Eu(NO3)3-6H2O, and In(NO3)2-6H2O would decompose to form the mixed solutions of Zn2+ and In3+ ions or Zn2+ and Eu3+ ions in DI water. It is impossible for In3+ and Eu3+ ions to be separated with Zn2+ ions from the solutions during the growth processes of ZnO-based nanowires, the Eu-doped and In-doped ZnO nanowires could be grown from the mixed solutions.

Normally, when trivalent ions are introduced into the wurtzite structure, some changes in the lattice parameters are observed since the dopants ions are bigger or smaller and there is a charge imbalance that has to be overcome producing defects and distortions in the structure. The peaks in the XRD do not shift in any of the samples, no matter if the dopant ion is Eu or In and the amount of it. The authors do not give any indication of the value of the lattice parameters. They describe changes in the FWHM and intensity of the peaks but no specific data are given and it is not possible to see these changes from the figures they show.

Answer: Thanks for reviewer’s comment. Because the shifts of 2θ values in the Eu3+-doped and In3+-doped ZnO nanowires are unapparent, we do not to discuss their variations. We re-checked the XRD patterns and well described the variations of 2θ and FWHM values of the (200) diffraction peak. Please see lines 258-267, 277-280, and 283-288.

They also claim that since in some of the FESEM images the nanowires seem to be not regular hexagons the dopants have to be incorporated but, in my opinion, this is not enough reason for this conclusion. In fact, how many FESEM images have they taken? Have they made any statistics of this change of shape? In particular in figure 4, part c, it seems to me that the shape change is an artifact of the image due to thermal drift during image acquisition. No compositional analysis of any type has been performed, so it is not clear that any of the dopants are in fact incorporated into the ZnO nanowires.

Answer: Thanks for reviewer’s comment. At least five images were measured for each Eu3+-doped and In3+-doped ZnO nanomaterials to confirm their surfaces’ morphologies, and the five data had the same results, please see the attached SEM image below. The EDS analyses were added in table 1 to prove the Eu3+ and In3+ ions incorporated into ZnO-based nanowires, please see lines 241-250.

Regarding the luminescence experiments, they are poorly explained and discussed. There is no explanation of why a wavelength of 393 nm has been used for PLE spectra.

Answer: Thanks for reviewer’s comment. We used a wavelength of 395 nm for PLE spectra, the error was corrected, please see line 310.

The authors do not really explain why there are differences between the spectrum performed on undoped and doped samples.

Answer: Thanks for reviewer’s comment. We explained why there are differences between the spectrum performed on undoped and doped samples, please see lines 401-411.

In the PL spectra, they claim that there is a change from 395 to 396 and 397 in the emission peak position, but 1 nm resolution is under the possibilities of the technique at room temperature.

   Answer: Thanks for reviewer’s comment. About the “a change from 395 to 396 and 397 in the emission peak position”, we edited the description for your query, please see lines 339-343.

The wavelengths resulting in the maximum emission intensities were 395 ± 2, 396± 2, and 397 ± 2 nm for 100°C-grown undoped ZnO, 60°C-grown ZnO-40-Eu, and 88°C-grown ZnO-4-In nanowires, respectively. These results suggest that the additions of Eu3+ or In3+ affect the maximum intensities of the emission spectra but they cannot almost affect wavelengths resulting in the maximum emission intensities.

In addition, the results discussion is limited to a list of different results of various papers and no comparison with their results is made.

   Answer: Thanks for reviewer’s comment, we added the comparison in lines 383-391.

It is surprising that no emissions related to Eu are observed since this is one of the most efficient luminescent ions.

   Answer: Thanks for reviewer’s comment. The emission related to 60°C-grown ZnO-40-Eu nnaowires was shown in Figure 8.

Reviewer 3 Report

The manuscript represents the photoluminescence properties of ZnO-Based nanowires using hydrothermal methods. The authors deposited 200 nm ZnO film on SiO2/Si substrates as the seed layer. Then doped Eu and In in the structure to have Eu-doped and In-doped ZnO nanowires at different synthesis temperatures. They investigated the effect of different amounts of doping of Eu and In on ZnO nanowires. The authors doped ZnO nanowires using the hydrothermal method at low temperature (below 100oC) and investigate the effects of different concentrations of Eu3+ and In3+ ions on their crystalline and photoluminescence properties.

The experiments are organized well, and the discussion is supportive of the results.

The value of the paper is enough to be published in this journal.

Author Response

Reviewer 3:

   The manuscript represents the photoluminescence properties of ZnO-Based nanowires using hydrothermal methods. The authors deposited 200 nm ZnO film on SiO2/Si substrates as the seed layer. Then doped Eu and In in the structure to have Eu-doped and In-doped ZnO nanowires at different synthesis temperatures. They investigated the effect of different amounts of doping of Eu and In on ZnO nanowires. The authors doped ZnO nanowires using the hydrothermal method at low temperature (below 100oC) and investigate the effects of different concentrations of Eu3+ and In3+ ions on their crystalline and photoluminescence properties. The experiments are organized well, and the discussion is supportive of the results. The value of the paper is enough to be published in this journal.

Answer: Thanks for reviewer’s agreement to our research.

Round 2

Reviewer 2 Report

After carefully reading the revised version of the manuscript, I find that the authors have only partially addressed the questions and comments made by this referee. In particular, they have only included EDX measurements and discussed some aspects related to the crystal quality of the samples. Nevertheless, there is still a lack of proper analysis of XRD results. In addition, the authors talk about Eu3+ doped  and In3+ doped samples, which is absolutely speculative, since EDX is not sensible to the oxidation state of the dopant. Actually, the absence of the characteristic Eu3+ red emission could be due to the presence of Eu2+ ions, with enhanced emission in the blue-green range of the visible spectrum.

The novelty of this work is, in my opinion, marginal. Hence I regret to say I cannot recommend this paper for publication.

Author Response

Second revision:

After carefully reading the revised version of the manuscript, I find that the authors have only partially addressed the questions and comments made by this referee. In particular, they have only included EDX measurements and discussed some aspects related to the crystal quality of the samples. Nevertheless, there is still a lack of proper analysis of XRD results. In addition, the authors talk about Eu3+ doped and In3+ doped samples, which is absolutely speculative, since EDX is not sensible to the oxidation state of the dopant. Actually, the absence of the characteristic Eu3+ red emission could be due to the presence of Eu2+ ions, with enhanced emission in the blue-green range of the visible spectrum.

Answer: The XRD’s analyses for your requirements were added in lines 258-267, 277-280, and 283-288 please check them.

However, I need to say that the XRD cannot be used to find the dopants have into the ZnO nanowires or not. EDS analysis has good resolution for different metal in the oxidation state.

You said “the absence of the characteristic Eu3+ red emission could be due to the presence of Eu2+ ions, with enhanced emission in the blue-green range of the visible spectrum.”, I do not think it is correct. Because the Eu2+ ions are not a stable condition and Eu3+ ions are in a stable state as Eu2O3 is used as dopant material. Many researches in phosphors show that if you want to reduce the Eu3+ ions into Eu2+ ions, you need to heat them in a reducing atmosphere in high temperature. Many researches we mentioned in this paper cannot confirm the reason the cause Eu-doped and In-doped ZnO nanowires having different PL spectra, please References 17-18 and 23-29. But in this study we have really found the reason to PL spectra of Eu-doped and In-doped ZnO nanowires, please see lines 401-412.

The novelty of this work is, in my opinion, marginal. Hence I regret to say I cannot recommend this paper for publication.

However, I need to say that so far no any researches use Zn(NO3)2-6H2O, C6H12N4, and Eu(NO3)3-6H2O or In(NO3)2-6H2O to investigate the Eu-doped and In-doped ZnO nanowires and investigate their growth properties under different concentrations of ions. Also, no any researches prove t the concentrations of Eu3+ and In3+ ions will affect the growth temperatures of Eu-doped and In-doped ZnO nanowires and investigate the effect of the concentrations of Eu3+ and In3+ ions on the growth morphologies and PL properties. Please see lines 114-117.